# The Wavelength-Based Inactivation Effects of a Light-Emitting Diode Module on Indoor Microorganisms

**DOI:** 10.3390/ijerph19159659

**Published:** 2022-08-05

**Authors:** Jong-Il Bang, Ji-Hi Kim, Anseop Choi, Minki Sung

**Affiliations:** 1Department of Architectural Engineering, Sejong University, 209 Neungdong-Ro, Gwangjin-Gu, Seoul 05006, Korea; 2Specialization Strategy Technology Department, EAN Technology Co., Ltd., EAN Institute of Sustainable Technology, 77-gil Teheran-Ro, Gangnam-Gu, Seoul 06159, Korea

**Keywords:** wavelength (275; 370; 385; 405 nm), light-emitting diode (LED), inactivation effect, microorganisms

## Abstract

With the increased incidence of infectious disease outbreaks in recent years such as the COVID-19 pandemic, related research is being conducted on the need to prevent their spread; it is also necessary to develop more general physical–chemical control methods to manage them. Consequently, research has been carried out on light-emitting diodes (LEDs) as an effective means of light sterilization. In this study, the sterilization effects on four types of representative bacteria and mold that occur indoors, *Bacillus subtilis*, *Escherichia coli*, *Penicillium chrysogenum*, and *Cladosporium cladosporidides*, were confirmed using LED modules (with wavelengths of 275, 370, 385, and 405 nm). Additionally, power consumption was compared by calculating the time required for 99.9% sterilization of each microorganism. The results showed that the sterilization effect was high, in the order 275, 370, 385, and 405 nm. The sterilization effects at 385 and 405 nm were observed to be similar. Furthermore, when comparing the power consumption required for 99.9% sterilization of each microorganism, the 275 nm LED module required significantly less power than those of other wavelengths. However, at 405 nm, the power consumption required for 99.9% sterilization was less than that at 370 nm; that is, it was more efficient and similar to or less than that at 385 nm. Additionally, because 405 nm can be applied as general lighting, it was considered to have wider applicability and utility compared with UV wavelengths. Consequently, it should be possible to respond to infectious diseases in the environment using LEDs with visible light wavelengths.

## 1. Introduction

Following the severe acute respiratory syndrome (SARS) and Middle East respiratory syndrome (MERS) outbreaks, which caused many deaths worldwide, 2019 was the beginning of the COVID-19 epidemic, which is still evolving. As recently reported, COVID-19 is transmitted by droplets from an infected person. Moreover, such infections occur more easily indoors than outdoors under the same conditions. Therefore, caution is required to prevent the spread of this and other infectious diseases in automobiles, public transportation, commercial facilities, and medical facilities managing infected patients, which are used by many people at a time. As such, there is a fundamental need for developing a means of physically and chemically controlling infectious agents [1,2].

To control and manage the microorganisms such as viruses that cause these infectious diseases, ultraviolet germicidal irradiation (UVGI) can be used because it is known to be effective in sterilizing microorganisms [3]. UVGI is a method of controlling the growth and reproduction of microorganisms by irradiating UV-C rays directly onto the target surface on which microorganisms are distributed. Even short periods of UV irradiation have a strong sterilization effect. Usually, UVGI is used as a UV lamp; however, it can be difficult to use these lamps in confined spaces owing to size limitations. Accordingly, research and development on sterilization using smaller light-emitting diodes (LEDs) has been conducted. LEDs are energy efficient and have a long lifespan. Moreover, there are fewer application restrictions owing to their small size than those for UVGI lamps. LEDs can also be easily used in everyday life and in various industrial fields. However, when humans are directly exposed to UV-C light, it can have harmful effects, causing eye damage, aging skin, and increased risk of skin cancer, which limits its use. Moreover, if specific materials are exposed to UV-C light for more than a certain period, they may be damaged owing to structural deterioration. For this reason, UVGI is best used in applications that minimize human exposure, making it difficult to apply them in a general living environment.

Upper room (UR)-UVGI sterilizes the air circulating in the upper spaces in rooms, avoiding any exposure to people in the room; it is commonly used in medical facilities. Sung et al. [4] tested the sterilization effect of UR-UVGI in a room in the general ward of a hospital, analyzing the sterilization effect based on the ventilation efficiency using computational fluid dynamics (CFD). Bang et al. [5] experimentally verified the sterilization effect of UR-UVGI in a negative pressure isolation ward—a facility to isolate infected patients—and compared the sterilization effects based on the installation location using CFD. In-duct (ID)-UVGI, which is installed and used inside an air handling unit (AHU), is easy to use because people are not directly exposed to it. However, an appropriate lamp should be selected based on the size or type of the AHU. Sung et al. [6] investigated the contamination of the evaporative humidification element in a hospital AHU and confirmed that the concentrations of airborne and surface microorganisms had been reduced through long-term ID-UVGI. They also confirmed that the contamination of the element surface was reduced. Bang et al. [7] investigated the contamination of the element in the direct evaporative cooling process of a liquid dehumidifying AHU. ID-UVGI was then installed, and the sterilization effect of microorganisms on the surface and cooling water was confirmed. Standardized UV lamps are currently used in this format, but only for sterilization in specific facilities.

As discussed above, to replace UV mercury lamps, research has been conducted to develop UV–LEDs [8] that can deliver sterilization and target specific microorganisms. Malik et al. [9] found that 385 nm UV-A LEDs were effective in sterilizing *Escherichia coli* (irradiation for 1 h), thereby offering great potential in the medical field. Kim and Kang [10] evaluated the performance of UV-C LEDs in air sterilization by generating airborne microorganisms in an experimental chamber. Both bacteria and mold were effectively sterilized, suggesting that UV–LEDs could replace mercury lamps. The use of mercury is restricted per the Minamata Convention, and UV–LEDs are expected to replace UV mercury lamps based on the results in [10]. As such, research on air sterilization, UV–LED output, and other sterilization methods using UV–LEDs should be continued. Lai and Nunayon [11] used UV-C LEDs to sterilize toilet bowls. The targets were *E. coli*, *Salmonella typhimurium*, and *Staphylococcus epidermidis*. The peak wavelength of the UV-C LED was 269.6 nm. Different numbers of LEDs of various shapes were configured and tested. The sterilization effect based on increasing the number of simple LED devices slightly increased the air sterilization. Moreover, the authors established that a better sterilization effect could be achieved based on the arrangement of LEDs rather than their number. This sterilization method was considered to have remarkable potential because it was sustainable and localized in a compact form. Nunayon et al. [12] compared the sterilization effect of UR-UVGI with a mercury lamp (18.95 μW/cm^2^) and combined with a UV–LED (40.77 μW/cm^2^). After spraying *E. coli*, *Serratia marcescens*, and *S. epidermidis* into a space, inactivation experiments were conducted. The mercury lamp and UV–LED showed similar performances, with the LED showing different performance and energy consumption depending on the combination of LED devices. In particular, LEDs should be used with sufficient input power to produce high output. Therefore, there remains a need for research on the use of UV–LEDs.

As aforementioned, because UV–LEDs have a specific wavelength in the UV range, the sterilization effect can be obtained. However, it can be difficult to secure a sufficiently high output. Moreover, the use of UV–LEDs is limited owing to their potential health risk and cost. Accordingly, studies on sterilization using LEDs of visible light and UV-A wavelengths close to visible light are being conducted. Studies in various fields have been conducted on the sterilization effect of 405 nm wavelength in visible light. Murdoch et al. [13] used 405 nm LEDs to evaluate the sterilization effect on *Campylobacter jejuni* and Gram-negative bacteria. It was suggested that it could be used effectively using the stability of visible light (non-UV rays) and a high intensity of 405 nm. Sommers et al. [14] conducted a sterilization experiment using 405 nm visible light treatment in chicken processing that they found to be highly effective.

For pathogens, Murdoch et al. [15] evaluated the sterilization effect based on irradiation time on the bacterial pathogens *Salmonella, Shigella*, *E. coli*, *Listeria*, and *Mycobacterium* (under surface and dilution conditions). The sterilization effect of the non-UV wavelength at 405 nm was considered to be broadly applicable, given that it is safe for humans. A study on the photodynamic inactivation of microorganisms using visible light wavelengths that were not harmful to mammalian cells was conducted. Additionally, the sterilization effect against COVID-19 was also evaluated [16].

As the interest in sterilization through visible light increased in the photochemistry and biology fields, a sensitivity test for *Listeria monocytogenes* was conducted. The 400–450 nm range was compared at intervals of 10 nm. At a sufficiently high dose, the highest inactivation effect could be observed at 405 ± 5 nm [17]. Li et al. and Kumar et al. [18,19] conducted an inactivation experiment based on the experimental progress of six bacteria using 405 nm at room, low, and cryogenic temperatures (25, 10, 15, and 4 °C). They considered that there was a more efficient LED combination model for each type of bacteria. The 405 nm LED showed significant potential. Sklar et al. [20] compared the sterilization effects of UV, visible light, and infrared light and the effects on human skin cells based on the wavelengths being compared. Guffey et al. [21] confirmed the sterilization effects of visible light at 405 and 470 nm against *S. aureus* and *Pseudomonas*.

Blue light is known to exhibit an effect similar to UV light, but it was confirmed that it differed depending on the type of microorganism. Murdoch et al. [22] evaluated the inactivation effects on yeast and mold using high-intensity 405 nm light experiments conducted on *Saccharomyces cerevisiae*, *Candida albicans*, and *Aspergillus niger*. The results showed that the chemical reaction of mold spores and inactivation at 405 nm was effective in mold decontamination.

As shown in Table 1, most of the sterilization effects of wavelengths in the visible and UV ranges were compared [18,23,24,25,26]. The sensitivity coefficient for microorganisms using UV-C can be derived from the dose and survival rate and was used to evaluate or predict the sterilization effect [3]. However, although previous research analyzed the sterilization effects based on the type of microorganism and the wavelength of light, most of the experiments were conducted over a short distance of approximately 5 cm. To apply sterilization to a general environment, it is necessary to examine the sterilization effect over a longer distance. In addition, the sterilization effect was confirmed for each wavelength, but there was no comparison of energy consumption and quantitative sterilization effect for each wavelength to obtain the target sterilization under the same experimental conditions. Most prior studies have verified that visible light has a sterilizing effect on microorganisms, but it is necessary to compare the quantitative sterilization effect efficiency with other wavelengths. In this study, the sterilization effects of 275, 370, 385, and 405 nm wavelength LEDs were confirmed at a distance of approximately 15 cm (for two types of bacteria and mold). In addition, the energy required for 99.9% sterilization was calculated, and the efficiency of each wavelength was compared with respect to the energy consumption. The sterilization effects of visible light and power consumption were compared with the sterilization effects of the near-UV (UV-A) and UV-C wavelengths to demonstrate its potential application. Overall, this study aimed to provide results that can be used to apply the visible light sterilization in general environments.

## 2. Materials and Methods

### 2.1. LED Module

The LED module used in this experiment was manufactured to deliver a specific output at a specific wavelength. The LED array was attached to a heat sink and a fan (see Figure 1a) to minimize the module’s temperature and prevent thermal effects from interfering with the experiment. A frame (15.5 × 11.5 cm) covered the LED module including the LED array, heat sink, and fan (see Figure 1b). The height of the frame was 15 cm from the LED array. UV-A (385 nm) and visible light (405 nm) LED devices (Hontiey, Shenzhen, China), each with an output of 3 W, were arranged in a 4 × 3 array for each module, as in Figure 1d. An LED strip was added to form a 4 × 4 array for the 370 nm module because the output of the 370 nm LED was comparatively low. The UV-C (275 nm) LED (Shenzhen Santang Lighting Co., Ltd., Shenzhen, China) had 5 LEDs connected in a line (0.4 W), and 3 lines were arranged in a 5 × 3 array, as shown in Figure 1c. The rated voltage of all modules was 12 V.

The intensities of the LED modules were confirmed based on their wavelength characteristics and irradiation distances using a spectrometer (Black-Comet C-50, StellarNet, Tampa, FL, USA), the specifications of which are listed in Table 2. The intensity was measured at 5 points at a distance of 15 cm from the LED module, as shown in Figure 2b, after which the average was used. The intensity was measured after using an agar petri dish cover. Spectrometer measurements confirmed that 96–98% of the light at 370, 385, and 405 nm penetrated the agar petri dish cover. For 275 nm, the light rarely penetrated the plastic cover; thus, a separate quartz glass plate was used as a petri dish cover. The quartz glass plate exhibited a transmittance of approximately 90% for UV-C (275 nm).

### 2.2. Microorganisms

Researchers have conducted experiments on wavelength-based sterilization effects on various microbial species including bacteria, viruses, and mold. In this study, two types of bacteria and two types of molds were tested. The bacteria selected were *E. coli* (KCTC 22002), which is mainly present in environments contaminated by feces or in the food industry, and *Bacillus subtilis* (KCCM 11316), a Gram-positive bacterium belonging to the *Bacillus genus* that is widely distributed in general living environments including soil, the human body, and animals. For mold, experiments were conducted on *Cladosporium cladosporidides* (KCTC 26803), which is mainly detected in residential, indoor, and outdoor environments, and *Penicillium chrysogenum* (KCTC 6933), a type of green mold. All biological resources used in this research were obtained from the Korean Culture Center of Microorganisms (KCCM) and Korean Collection for Type Cultures (KCTC).

### 2.3. Experiment Setup

The experimental cases are listed in Table 3. The irradiation times for the 275, 370, 385, and 405 nm LED modules were set differently based on the expected sterilization effect of each microorganism being examined. The 275 nm irradiation time was shorter because the sterilization effect of the UV-C rays was higher. The set times were 2 and 10 min for the bacteria and mold, respectively. The irradiation times of the 370, 385, and 405 nm sources were set to 15 min intervals for the bacteria and 30 min intervals for the mold [3,9,18,23,24,25,26].

The LED irradiation experiment was conducted as follows: The microorganisms were diluted from a lyophilized strain in a NaCl (0.85%, saline) 9 mL sterile solution. The dilutions were 10^–4^ times for *B. subtilis* and 10^–3^ times for *E. coli*, *C. cladosporidides*, and *P. chrysogenum*. The dilution step was calculated in advance based on the confirmation of the volume of microorganisms detected according to the dilution concentrations. Each diluted solution had 0.3 mL extracted, with the bacteria being dispensed on trypticase soy agar (TSA) and the mold on potato dextrose agar (PDA), before being evenly applied using a scraper. After the application of the microorganisms, they were irradiated by the 370, 385, and 405 nm LEDs and the 275 nm LED (with light from the LED module). The specimens were appropriately covered to prevent the agar petri dish and strain solution from drying out during the experiment; using intensity measurements of the LED module, it was established that the quartz glass and agar petri dish covers had no significant effect on the irradiating LED light. In addition, using T-type thermocouples and a GL820 Data Logger (Graphtec Corporation, Yokohama, Japan), the agar petri dish surface temperatures (based on operation of the LED module) were also measured. At this time, the experiment was conducted in a bio-clean bench. The experiments with the 370, 385 and 405 nm modules were conducted simultaneously, and that for 275 nm was conducted alone. Not only the surface temperature but also the air temperature and humidity inside the bio-clean bench were measured. The experimental bacteria were incubated at 32 °C for 1–2 days and the experimental mold at 25 °C for 4–5 days [27,28]. After incubation, the colony-forming units (CFUs) on the agar petri dish were counted.

### 2.4. Survival Rate

The survival rates (SRs) can be determined based on calculations. In general, sterilization using irradiating rays can be expressed as follows [3]:(1)$$SR=e^{-kIt}=\frac{N_{t}}{N_{0}}$$
where *k* is the sterilization coefficient (m^2^/J), *I* is the intensity (W/m^2^), *t* is the exposure time (s)—that is, the sterilization effect is proportional to the sterilization coefficient, intensity, and exposure time—*N_t_* is the microorganism concentration level after *t* s irradiation, and *N_0_* is the initial concentration. The term “dose,” that is, the product of intensity (*I*) and irradiation time (*t*), can be thought of as the amount of irradiation received by the microorganism. After substituting the experimental results into Equation (1), *k* can be calculated based on the wavelength for each microorganism using the logarithmic *SR* and dose. The sterilization coefficient (*k*) indicates the characteristic of each microorganism and how susceptible to UV the microorganisms is. Generally, it means that a large *k* indicates higher sterilization performance. In this study, coefficients not only for UV but also for other wavelengths were calculated and compared.

## 3. Results

### 3.1. Thermal Conditions

A sterilization experiment for each LED module was conducted on a bio-clean bench. It was confirmed that there was no microbial contamination inside the bench using UV-C sterilization before the experiment began. At 370, 385, and 405 nm, the temperature and relative humidity within the bio-clean bench during the *B. subtilis* experiment were 24.7 °C and 58%, respectively, before the LED operation and 30 °C and 48%, respectively, after operation. The temperature and relative humidity during the *E. coli* experiment were 24.7 °C, 57%, respectively, before the LED operation and 30.1 °C, 48%, respectively, after operation. During the *P. chrysogenum* experiment, the temperature and relative humidity were 25 °C and 61%, respectively, before the LED operation and 30.1 °C and 50%, respectively, after operation. For the *C. cladosporidides* experiment, the temperature and relative humidity were 23.4 °C and 52%, respectively, before the LED operation and 28 °C and 51.6%, respectively, after operation. At 275 nm, the temperature and relative humidity were 26.8 °C and 31%, respectively, before the LED operation and 27.2 °C and 30.8%, respectively, after operation in the *B. subtilis* experiment. For the *E. coli* experiment, the temperature and relative humidity were 27.1 °C and 24.6%, respectively, before the LED operation and 27.3 °C and 24.1% respectively, after operation. For the *P. chrysogenum* experiment, the temperature and relative humidity were 24 °C and 36%, respectively, before the LED operation and 25.5 °C and 33.3%, respectively, after operation. For *C. cladosporidides*, the temperature and relative humidity were 22.9 °C and 46.1%, respectively, before the LED operation and 25 °C and 42%, respectively, after operation. Since 370, 385 and 405 nm were measured at the same time, the temperature rise was higher than that of 275 nm alone. Considering the temperature rise for each module, it seems that the temperature rise by the module is not significant.

The surface temperature was measured separately using T-type thermocouples in the TSA and PDA petri dishes after operating the LED module for up to 2 h (the maximum irradiation time). The surface temperature results were as follows: At 275 nm, the surface temperature was 18.1 °C before the LED operation and 22.1 °C after operation. At 370 nm, it was 19.9 °C before the LED operation and 26 °C after operation. At 385 nm, it was 20.7 °C before the LED operation and 26.8 °C after operation. At 405 nm, it was 21.2 °C before the LED operation and 26 °C after operation. At 275 nm, the surface temperature increase was low when compared with that at the other wavelengths.

### 3.2. Inactivation Effect

#### 3.2.1. Reduction Rate

Figure 3 shows a comparison of the reduction rates based on the type of microorganism for each wavelength, with 370, 385, and 405 nm showing the highest sterilization effects in *E. coli*. Overall, the sterilization effect on bacteria was higher than that on mold. At 370 and 385 nm, the reduction rate of *B. subtilis* increased with irradiation time; after 45 min of irradiation, the reduction rates were 62% and 71%, respectively, and the reduction rate did not change significantly. At 405 nm, the reduction rate gradually increased with irradiation time. Lastly, the reduction rate was like that at 370 and 385 nm. The reduction rate for *E. coli* was 63% at 370 nm irradiation for 15 min, with 385 and 405 nm showing a reduction rate of 80% or more after 15 min of irradiation. In addition, the 370 nm wavelength showed a 99% reduction after 45 min of irradiation, with 385 and 405 nm showing a 98% reduction after irradiation of 30 min or longer.

For *P. chrysogenum* and *C. cladosporidides*, the reduction rate gradually increased with irradiation time at 370, 385, and 405 nm. *P. chrysogenum* showed a reduction rate of 50% or more when irradiated for more than 90 min. Finally, both *P. chrysogenum* and *C. cladosporidides* only showed a reduction rate of approximately 55%, even after 120 min of irradiation. The 370, 385, and 405 nm cases were compared at the same irradiation times with the 275 nm case being tested at a short irradiation time, as reported in previous studies. Compared with 370, 385, and 405 nm cases, the 275 nm case showed a higher reduction rate with only short irradiation times. In the case of *B. subtilis*, irradiating for approximately 2 min showed a reduction rate of 74%, exhibiting a reduction effect after 1 h irradiation with a different wavelength. In the case of *E. coli*, a reduction rate of 91% was achieved with only 20 s of irradiation. The 275 nm wavelength also showed a higher reduction rate of bacteria compared with that of the mold, like the other wavelengths. *P. chrysogenum* showed a reduction rate of 54% after 3 min of irradiation, showing a similar sterilization effect to that of the other wavelengths after 2 h of irradiation. Finally, after 7 min of irradiation, the reduction rate was 98%. *C. cladosporidides* showed a reduction rate of 72% after irradiation for 5 min, exhibiting a higher reduction rate than irradiation with other wavelengths for 2 h. A reduction rate of 93% was achieved after only 8 min of irradiation.

Statistical analysis using one-way ANOVA shows that there were differences (*p* < 0.05) in the reduction rate by the irradiation time for all wavelengths and microorganisms except for 370 nm with *P. chrysogenum* (*p* = 0.09). It was impossible to compare the reduction rates by wavelength and microorganisms because the irradiation times were different between them. Moreover, it is not appropriate to compare the sterilization performance only by comparing the reduction rate because the intensities of the LED modules for each wavelength were significantly different from each other in the experiments. Therefore, it is necessary to compare the sterilization performance through the concept of the dose of the 2.4 survival rate. This can be evaluated by deriving the sterilization coefficient *(**k*) for the wavelength of each microorganism.

#### 3.2.2. Sterilization Coefficient, *k*

Table 4 shows the effects of irradiation time on *B. subtilis* inactivation at each wavelength. Figure 4 shows the plots of the *SR* of B. subtilis against the dose. The shorter wavelength is more rapidly inactivated than the longer wavelength, and it can be confirmed that higher *k* has higher sterilization performance. The *k* values were as follows: at 275 nm: 0.021 m^2^/J, at 370 nm: 0.508 × 10^−4^ m^2^/J, at 385 nm: 0.231 × 10^−4^ m^2^/J, and at 405 nm: 0.182 × 10^−4^ m^2^/J. The *k* value at 275 nm was similar to that shown in a study by Kowalski [3] (Rentschler 1941, Munakata 1975, Munakata 1972).

Table 5 shows the effects of irradiation time on *E. coli* inactivation for each wavelength. Figure 5 shows the *SR* of *E. coli* based on the dose. The *k* values were as follows: at 275 nm: 0.235 m^2^/J, at 370 nm: 2.566 × 10^−4^ m^2^/J, at 385 nm: 1.417×10^−4^ m^2^/J, and at 405 nm: 1.017 × 10^−4^ m^2^/J. At 385 and 405 nm, approximately 100% sterilization was achieved when irradiated for 1 h. The *k* value at 275 nm was like that shown in a study by Kowalski [3] (Rentschler 1942, Quek 2008, Collins 1971).

Table 6 shows the effects of irradiation time on the inactivation of *P. chrysogenum*. Figure 6 shows the *SR* of *P. chrysogenum* based on the dose. The *k* values were as follows: at 275 nm: 0.014 m^2^/J, at 370 nm: 0.195 × 10^−4^ m^2^/J, at 385 nm: 0.059 × 10^−4^ m^2^/J, and at 405 nm: 0.056 × 10^−4^ m^2^/J. The *k* value at 275 nm was like that shown in a study by Kowalski [3] (Luckiesh 1949).

Table 7 shows the inactivation effects of *C. cladosporidides* based on the irradiation time for each wavelength. Figure 7 shows the *SR* of *C. cladosporidides* based on the dose. The *k* values were as follows: at 275 nm: 0.011 m^2^/J, at 370 nm: 0.16 × 10^−4^ m^2^/J, at 385 nm: 0.051 × 10^−4^ m^2^/J, and at 405 nm: 0.042 × 10^−4^ m^2^/J. The *k* value at 275 nm was like that shown in a study by Kowalski [3] (Luckiesh 1949).

### 3.3. Comparing Energy Consumption

The time required for the 99.9% sterilization for each microorganism could be calculated based on *k* for the microorganisms for each wavelength, as presented in Section 3.2. The measured energy consumption of LED modules was 13.8 W for 275 nm, 8.05 W for 370 nm, 6.03 W for 385 nm, and 5.83 W for 405 nm, respectively. Figure 8 shows a comparison of the required power consumption when the LED module was operated for the time required to achieve the calculated 99.9% sterilization. At 275 nm, it was predicted that 99.9% sterilization could be achieved with a short irradiation time with the power consumption being predicted to be highest, that is, at 4.5 Wh for *C. cladosporidides*. At 370 nm, the power consumption required for the sterilization of all the microorganisms was predicted to be the highest. Although 370 nm corresponded to a wavelength in the UV-A range, the power consumption required to obtain 99.9% sterilization was higher than that for 385 and 405 nm. In addition, the output of 370 nm was weak when manufacturing the LED module, even though four LED chips were added; the output was lower than those of 385 and 405 nm modules, as shown in Table 1. This indicated that the power consumption of this module was significant. At 385 and 405 nm, the power consumption required for 99.9% sterilization was similar, and in the case of *P. chrysogenum*, 405 nm was observed as effective. Based on these results, it can be inferred that a wavelength of 405 nm is highly effective for use in sterilization technology.

## 4. Discussion

In previous studies [18,23,24,25,26], experiments to confirm the sterilization effect of visible light were conducted at short distances ranging from 2 to 5 cm. In addition, many other studies were conducted at distances shorter than 10 cm. A distance of 15 cm was set in this study to permit more practical, and therefore significant, results for application, commercialization, and technicalization in general living environments. In addition, irradiated targets are affected by LED heat generation at short distances. It is necessary to secure an appropriate distance to minimize LED heat generation for an irradiated target. Future researchers should also investigate sterilization at longer distances to support general lighting. Moreover, it was considered necessary to review the sterilization effects on actual furniture, walls, and many unspecified direct contacts with microorganisms (rather than just the agar petri dish). It should be noted that the results of this experiment may differ from those of sterilization in air or water in terms of surface microorganisms.

The sterilization effect of the 405 nm visible light LED was determined as follows. The *k* value was derived using the microorganism and wavelength experimental results, and 275 nm (UV-C) was similar to that previously shown [3]. It was assumed that the validity of the experiment in [3] was verified. The *k* value for UV-A, visible light conducted in the same way was considered to be reliable. Although the sterilization effect of UV-A is clearly lower than that of UV-C, it was considered because it can be used in a living environment for safety and applicability.

In this study, an experiment was conducted by fabricating an LED module. Because the module had an error range of ±5–10 nm—that is, it did not produce a 405 nm wavelength precisely, the centroid wavelength and output may have differed between the modules. Consequently, specialized modular manufacturing would be required to obtain accurate LED specifications. In addition, the output (intensity) of the LED differed depending on the input (power) [8,11,12], making it necessary to secure an appropriate output to obtain the required sterilization effect. In the case of the module manufactured in this study, an LED chip was added to ensure the requisite intensity of the 370 nm module, but it exhibited no significant difference. The power required for 99.9% sterilization was estimated based on the manufactured module. The 370 nm module was predicted to consume more power than the 385 and 405 nm modules. This was likely because of differences in the LED output despite the same rated power being used.

Consequently, it was necessary to consider the rated power required to secure an appropriate output based on each wavelength. Moreover, it was found that the differences in the sterilization effects and power consumption between the 385 and 405 nm cases, which exhibited similar output, were not significant. It was thought that sterilization using visible light could have a similar effect on the sterilization effect or energy consumption in long-term irradiation; that is, the application range of visible light sterilization could be considered to be wide, and its application potential significant. To date, there have been many studies comparing sterilization effects based on wavelength; however, there has been no verification through comparisons of power consumption. The results of this study should be helpful in the manufacturing of lighting fixtures for visible light sterilization.

When UV-C directly irradiates humans, it can have harmful effects, thus restricting its use. However, considering that visible light is harmless to humans at an appropriate irradiance [29] and can be applied as lighting in a general living environment [30], it would be possible to irradiate areas over a long period of time to obtain the required sterilization effect. However, there is a need for more research in relation to the safety of visible light on human skin [31]. Accordingly, more research on the exposure safety of wavelengths should be conducted and carefully considered. In addition, 405 nm LEDs are less expensive than UV LEDs and can be freely produced in various forms along with 3D prints, and based on the measured results of the agar petri dish surface temperature, it is likely that the application range would be wider even if irradiation were to be conducted over a long period. This is because the temperature increase was not significant. Consequently, visible light sterilization has a high potential for application in various fields.

## 5. Conclusions

In this study, the sterilization effects of an LED module at wavelengths of 275, 370, 385, and 405 nm were compared. The sterilization effect of 275 nm—corresponding to UV-C—was the highest, with no significant difference in the sterilization effects of the 370 and 385 nm (UV-A), and 405 nm (visible light) cases. Sterilization using visible light could be an acceptable solution when countermeasures are required in a general living environment, particularly with the ongoing pandemic. Although the sterilization effect may be lower than that of UV sterilization, visible light sterilization was considered to have potential applicability. The results of this study are summarized as follows:

(1)The sterilization effect at 405 nm was inferior to that of UV-C sterilization. However, when compared with the sterilization effects of 370 and 385 nm in the near-UV range, this difference was insignificant.(2)Visible light sterilization required long-term irradiation to achieve its purpose. However, since visible light can be used in a general living environment, it could be expected that direct sterilization would be possible in spaces where people live through the application of general lighting. However, it is necessary to proceed with research on the safety of 405 nm. Therefore, the application of the visible light wavelength to the general environment should be considered for an appropriate range.(3)In terms of energy consumption, no significant difference was observed between the sterilization using visible light and that using UV-A wavelengths. The power consumed when using the LED module for the time required to sterilize 99.9% of each microorganism was compared. The results showed that the sterilization effect and power consumption of 405 nm were similar to or less than those at 385 nm. This had a significant impact owing to the differences in the module output; however, when visible light sterilization was used, its development value remained high if its output was considered.(4)In terms of the sterilization of microorganisms, research has been conducted in the fields of biology, food, and medicine, and the process can be of great value for commercial use. Considering the safety of visible light, it can be a means of countering infectious diseases in engineering buildings, including private and public spaces where people spend most of their time.

This study demonstrated the applicability of sterilization using visible light. Although visible light was inferior in terms of sterilization effect and energy consumption compared with UV-C, it had better applicability compared with UV-A. In addition, because it was there was a comparison using only visible light, experiments and verification of various environments and actual infectious bacteria should be conducted through applications to actual lighting equipment.

## Figures and Tables

**Figure 1 ijerph-19-09659-f001:**
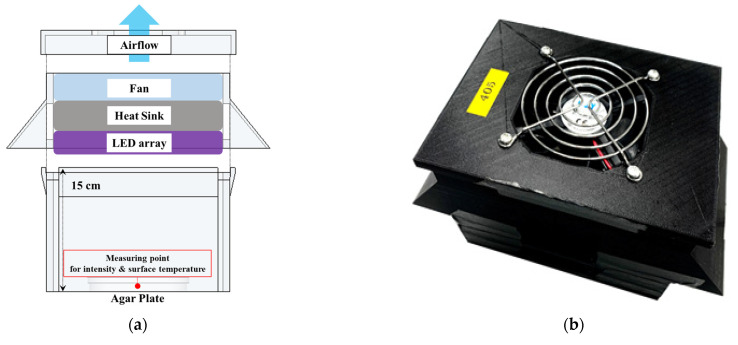
Components of the LED module used in this study. **(a)** The LED module concept diagram. (**b**) The actual module frame made with a 3D printer. (**c**) Array of the 275 nm LED attached to a heat sink. (**d**) Array of the 405 nm LED attached to a heat sink.

**Figure 2 ijerph-19-09659-f002:**
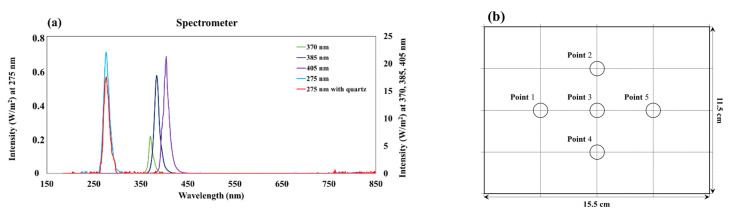
Specifications of the LED modules used in this study. (**a**) Spectra and intensities of the LED modules by wavelength. (**b**) The measurement points of LED module intensity.

**Figure 3 ijerph-19-09659-f003:**
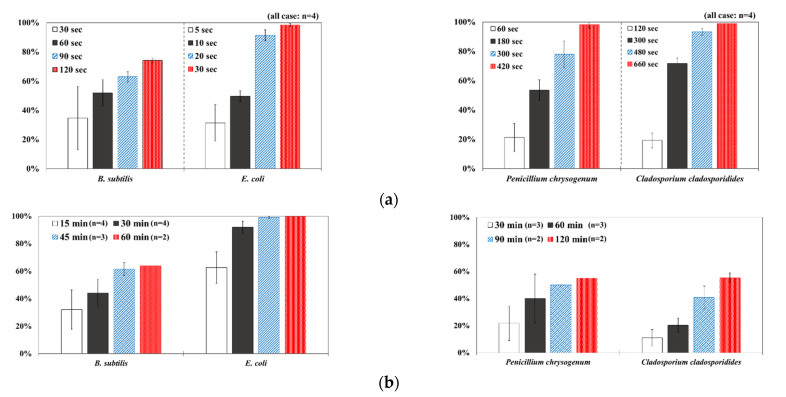
Inactivation rates (±std) of the four microorganisms by LED wavelength. (n indicates the number of replicates experiments.) (**a**) Reduction rate with microorganisms at 275 nm. (**b**) Reduction rate with microorganisms at 370 nm. (**c**) Reduction rate with microorganisms 385 nm. (**d**) Reduction rate with microorganisms 405 nm.

**Figure 4 ijerph-19-09659-f004:**
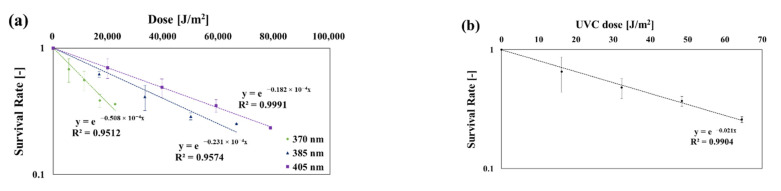
Inactivation curves of *B. subtilis* at each dose and wavelength: (**a**) 370, 385, and 405 nm; (**b**) 275 nm.

**Figure 5 ijerph-19-09659-f005:**
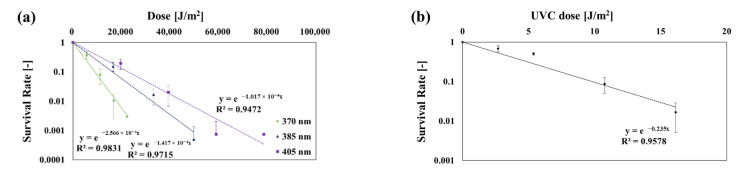
Inactivation curves of *E. coli* at each dose and wavelength: (**a**) 370, 385, and 405 nm; (**b**) 275 nm.

**Figure 6 ijerph-19-09659-f006:**
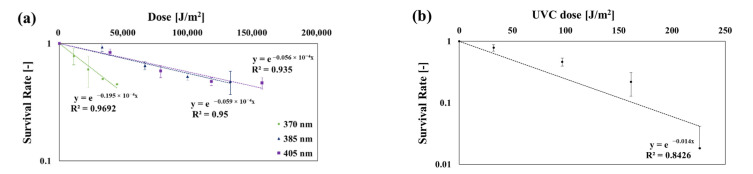
Inactivation curves of *P. chrysogenum* at each dose and wavelength: (**a**) 370, 385, and 405 nm; (**b**) 275 nm.

**Figure 7 ijerph-19-09659-f007:**
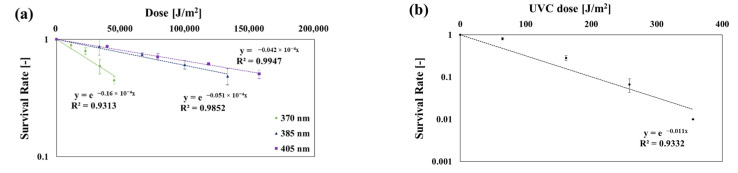
Inactivation curves of *C. cladosporidides* at each dose and wavelength: (**a**) 370, 385, 405 nm; (**b**) 275 nm.

**Figure 8 ijerph-19-09659-f008:**
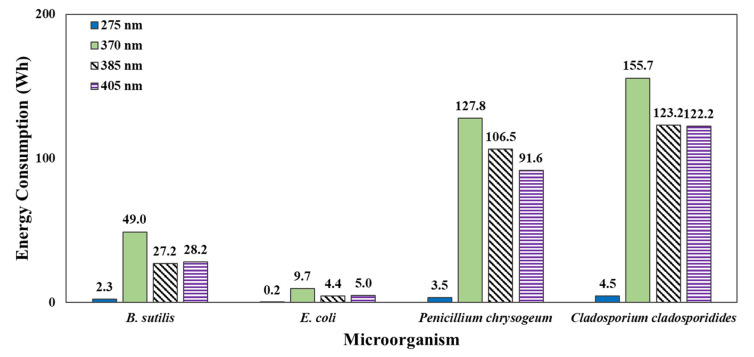
Energy consumption to achieve the 99.9% inactivation of four microorganisms by wavelength.

**Table 1 ijerph-19-09659-t001:** Th experimental conditions in previous research [18,23,24,25,26].

Microorganism	Temperature, °C	LED Wavelength, nm	Distance from Petri Dish, cm	Irradiation Time, h	Dose	Source
*Escherichia coli*, *Vibrio parahaemolyticus*, *Staphylococcus aureus*, *Salmonealla enterica*	4, 10, 20, 30, 40, 50	254, 365, 405	2	0.5	27 J/cm^2^	[23]
*S. aureus*, *Klebsiella pneumonia*, *Clotridium perfringerns*, *Acinetobacter aumannii*, etc. *(6)*	27 ±1	405	2	0.5, 1, 1.5, 2, 2.5, 3, 3.5, 4, 4.5, 5, 5.5, 6	10 mW/cm^2^	[24]
*E. coli*, *S. Typhimurium*, *Listeria monocytogenes*, *S. aureus*	10–15	461, 521, 642	4.5	7.5	22.1, 16, 25.4 mW/cm^2^	[25]
*E. coil*, *S. Typhimurium*, *Shigella sonnei*	4 ± 1	405 ± 5	4.5	7.5	18 mW/cm^2^	[26]
*L. monocytogenes*	4, 15, 25	405	4.5	8	26 ± 2 mW/cm^2^	[18]

**Table 2 ijerph-19-09659-t002:** Specifications of the UV-C, near-UV, and visible light LED modules.

	275 nm	370 nm	385 nm	405 nm
Peak wavelength (nm)	276.3	370.1	383.7	404.5
Centroid wavelength (nm)	277.1	372.7	385.6	405.7
FWHM (full width at half maximum)	12.81	9.58	12.07	12.65
Intensity (W/m^2^) at 15 cm	0.54 with quartz glass	6.2 with petri dish cover	18.4 with petri dish cover	21.8 with petri dish cover

**Table 3 ijerph-19-09659-t003:** Irradiation times by wavelength.

	*B. subtilis*	*E. coli*	*P. chrysogenum*	*C. cladosporidides*
275 nm ^a^	30 s	5 s	60 s	120 s
	60 s	10 s	180 s	300 s
	90 s	20 s	300 s	480 s
	120 s	30 s	420 s	660 s
370 nm 385 nm 405 nm	15 min ^a^		30 min ^b^	
	30 min ^a^		60 min ^b^	
	45 min ^b^		90 min ^c^	
	60 min ^c^		120 min ^c^	

Note: ^a^: *n* = 4, ^b^: *n* = 3, ^c^: *n* = 2.

**Table 4 ijerph-19-09659-t004:** Numbers of *B. subtilis* colonies by wavelength and irradiation time.

	Wavelength	275 nm	370 nm	385 nm	405 nm
Irradiation Time					
Control case 0 s		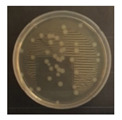	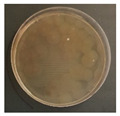 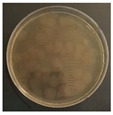		
CFU/mL		40.5	55.5		
275 nm: 30 s Other: 15 min		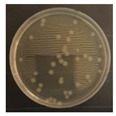	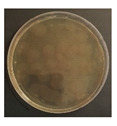	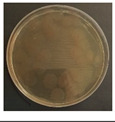	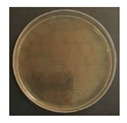
CFU/mL		26.5	37.75	34.5	38.88
275 nm: 60 s Other: 30 min		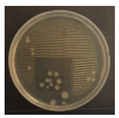	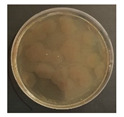	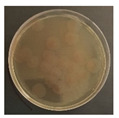	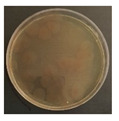
CFU/mL		19.5	31	23	27.13
275 nm: 90 s Other: 45 min		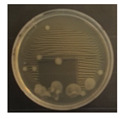	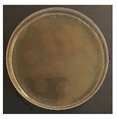	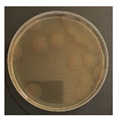	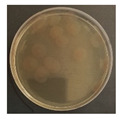
CFU/mL		15	21.33	16	19.5
275 nm: 120 s Other: 60 min		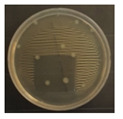	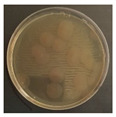	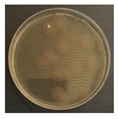	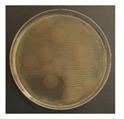
CFU/mL		10.5	20	14	13

**Table 5 ijerph-19-09659-t005:** Numbers of *E. coli* colonies by wavelength and irradiation time.

	Wavelength	275 nm	370 nm	385 nm	405 nm
Irradiation Time					
Control case 0 s		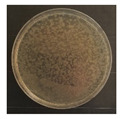	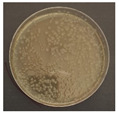 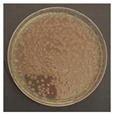		
CFU/mL		581	669		
275 nm: 5 s Other: 15 min		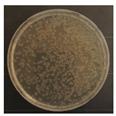	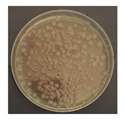	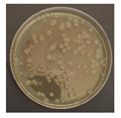	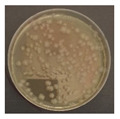
CFU/mL		398.5	249.5	102.75	130.63
275 nm: 10 s Other: 30 min		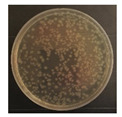	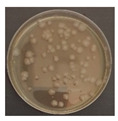	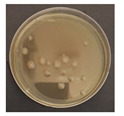	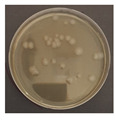
CFU/mL		293	54	11.25	13.38
275 nm: 20 s Other: 45 min		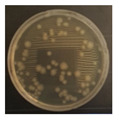	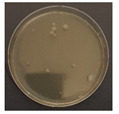	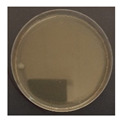	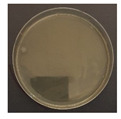
CFU/mL		50.75	7	0.3	0.5
275 nm: 30 s Other: 60 min		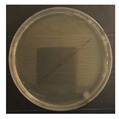	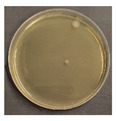	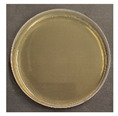	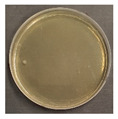
CFU/mL		9.75	2	0	0.5

**Table 6 ijerph-19-09659-t006:** Numbers of *P. chrysogenum* colonies by wavelength and irradiation time.

	Wavelength	275 nm	370 nm	385 nm	405 nm
Irradiation Time					
Control case 0 s		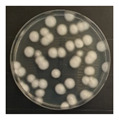	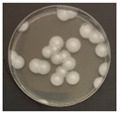 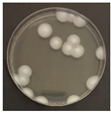		
CFU/mL		41	18.67		
275 nm: 1 min Other: 30 min		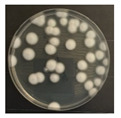	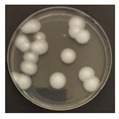	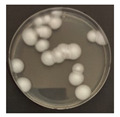	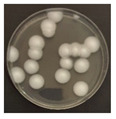
CFU/mL		32.3	15.67	18.67	16.83
275 nm: 3 min Other: 60 min		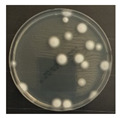	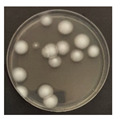	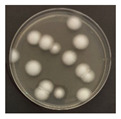	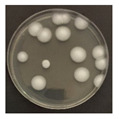
CFU/mL		19	12	13	11.67
275 nm: 5 min Other: 90 min		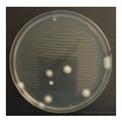	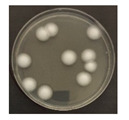	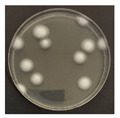	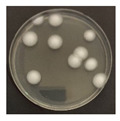
CFU/mL		9	10	10.5	9.5
275 nm: 7 min Other: 120 min		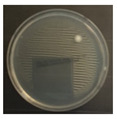	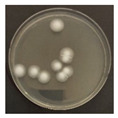	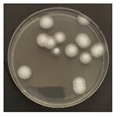	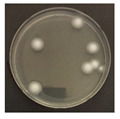
CFU/mL		0.8	9	9.5	9.3

**Table 7 ijerph-19-09659-t007:** Numbers of *C. cladosporidides* colonies by wavelength and irradiation time.

	Wavelength	275 nm	370 nm	385 nm	405 nm
Irradiation Time					
Control case 0 s		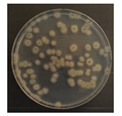	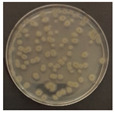 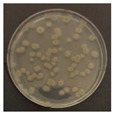		
CFU/mL		100.5	99.5		
275 nm: 2 min Other: 30 min		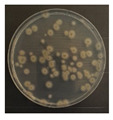	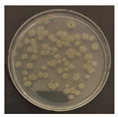	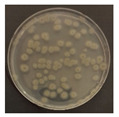	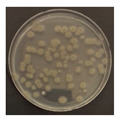
CFU/mL		81.25	88.67	87	86.83
275 nm: 5 min Other: 60 min		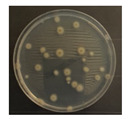	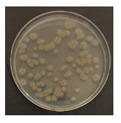	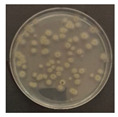	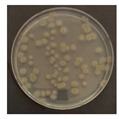
CFU/mL		28.5	79.33	74	70.5
275 nm: 8 min Other: 90 min		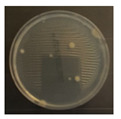	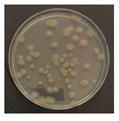	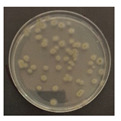	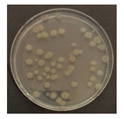
CFU/mL		6.75	59	60.5	61.5
275 nm: 11 min Other: 120 min		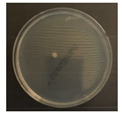	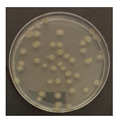	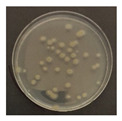	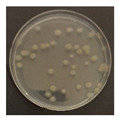
CFU/mL		1	44.5	48.5	50.5

## Data Availability

Not applicable.

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
