# Peer review of "The Wavelength-Based Inactivation Effects of a Light-Emitting Diode Module on Indoor Microorganisms"

_ijerph, 2022, doi:10.3390/ijerph19159659_

Round 1

Reviewer 1 Report

This article evaluated the sterilization effect of four microorganisms (2 bacteria and 2 mold) by LED lights with four wavelengths, one in the UVC, two in the UVA and one in the visible light, at a 15 cm distance. The required power of consumption of each model was evaluated.

This research is important, specially in replacing the lamps currently used with more functional LEDs. However, this manuscript and the methodology have some flaws and the aims need to be clearly stated. Is this manuscript testing the sterilization and power efficiency only? Or are the goals to modify the current practices of sterilization? If is it, how would the set up used in this manuscript would be efficient and used in real life sterilization applications (in air, for surface? What about the temperature increase caused by the set up?).

Introduction:

This could be improved. The table 1 is good, and I wish the text would reflect the table more. There is too many mention of papers without explaining their findings (wavelength – time – type of microorganism – efficiency observed) and comparing them to each other.

-       Ex: lines 59-61 – ‘’Sung et al. [4] tested the sterilization effect of UR-UVGI in a room in the general ward of a hospital, analyzing the sterilization effect based on the ventilation efficiency using computational fluid dynamics (CFD).’’ This sentence is useless without discussion of the findings.

Materials and methods:

-       Figure captions are all incomplete and do not contain enough details.

Ex: Figure1 – each panel needs to be explained.

What are the temperatures in Table 1 – temperature of the environment? If it is in the table, it should be discussed in the manuscript. What are the effect of temperature on sterilization, if any?

-       Table 3 – many treatments have a n of 2. Considering the methodology is not very complicated, this is not sufficient and should be replicated at least 3 times or more for statistical analysis.

-       Where are the statistical analysis? The is no test to compare survival rates between treatment and controls, and no test to compare the temperature increase. Statistical analyses are necessary before inferring any conclusions and comparisons of the treatments.

Results:

-       In this section also, the figure captions are incomplete and lack important details.

-       Temperature in the bench is increasing of almost 5°C after each sterilization process, this seems important. There is a 6°C increase in temperature at 370 and 385 nm, and a 5°C increase at 405 nm. This may create problems for the application in real life. Was this measured more than once?  In addition, the authors state that the surface increase is low at 275 nm compared to the other wavelength, is it lower, but it is still a 4°C increase, or 22%. This does not seem insignificant.

-       The n should be added in the figure captions.

-       Line 286-287 – this belongs in the discussion. Same lines 296-297, 306-307 and 315-316.

-       Figure 7 needs a Y axis title. What is the data presented, mean ± standard error or standard deviation? This needs to be in the figure caption, with the n.

Discussion and conclusions:

-       Globally, I think the discussion needs to be longer with comparison to the literature. There is a need to discuss the distance of sterilization (light to surface), this seem to be important in the introduction but is barely discussed and no comparison is made with literature.

-       Line 403: I do not agree with that statement, blue light, and visible light with wavelength close to the UV range are known to affect the human eye negatively. I agree that the effects of these wavelengths are much less important than the effects of UVC, however it would be false to assume they do not cause any damage to human. Hence, stating that they could be used in spaces with people in not appropriate. In this regard, I can not support the statement (2) of the conclusion.

-       Because aims were unclear in the introduction, I think the authors should rewrite them clearly and modify the discussion accordingly.

Author Response

We appreciate your kind comments. We have carefully read and responded to each comment and revised our manuscript.

This article evaluated the sterilization effect of four microorganisms (2 bacteria and 2 mold) by LED lights with four wavelengths, one in the UVC, two in the UVA and one in the visible light, at a 15 cm distance. The required power of consumption of each model was evaluated.

1) This research is important, specially in replacing the lamps currently used with more functional LEDs. However, this manuscript and the methodology have some flaws and the aims need to be clearly stated. Is this manuscript testing the sterilization and power efficiency only? Or are the goals to modify the current practices of sterilization? If is it, how would the set up used in this manuscript would be efficient and used in real life sterilization applications (in air, for surface? What about the temperature increase caused by the set up?).

Response:

1) First, the sterilization power of UV-A and visible light compared to that of UV-C was confirmed. In addition, the amount of power consumed to obtain the same sterilizing power was evaluated. As for the applications, this is an area for future related research. We believe that the results of this study can resolve the limitations of UV-C use (minimizing human exposure, degradation of materials, etc.) by suggesting the application of visible light LEDs in the general environment (both air and surface).

In actual sterilization applications, it is possible to install a visible light LED inside the heat exchanger (ventilation system). There are no additional concerns regarding degradation compared to UV-C, and from this study, we know that visible light LED is more effective in terms of energy consumption than UV-A when used for a long time. In addition, it can be manufactured in various forms; therefore, it can be freely applied in narrow spaces. Considering the applications in a general living environment, we judged that visible light LED can be used as general lighting.

Regarding the temperature rise, we have shown that LED heat generation can be mitigated by attaching a heat sink; therefore, this can be solved relatively easily depending on the application target.

Introduction:

2) This could be improved. The table 1 is good, and I wish the text would reflect the table more. There is too many mention of papers without explaining their findings (wavelength – time – type of microorganism – efficiency observed) and comparing them to each other.

Ex: lines 59-61 – ‘’Sung et al. [4] tested the sterilization effect of UR-UVGI in a room in the general ward of a hospital, analyzing the sterilization effect based on the ventilation efficiency using computational fluid dynamics (CFD).’’ This sentence is useless without discussion of the findings.

 Response:

2) There were many comments and the introduction read as a list; therefore, the introduction has been reduced and the comparisons of the theses were revised.

Materials and methods:

3) Figure captions are all incomplete and do not contain enough details.

Ex: Figure1 – each panel needs to be explained.

4) What are the temperatures in Table 1 – temperature of the environment? If it is in the table, it should be discussed in the manuscript. What are the effect of temperature on sterilization, if any?

5) Table 3 – many treatments have a n of 2. Considering the methodology is not very complicated, this is not sufficient and should be replicated at least 3 times or more for statistical analysis.

6) Where are the statistical analysis? The is no test to compare survival rates between treatment and controls, and no test to compare the temperature increase. Statistical analyses are necessary before inferring any conclusions and comparisons of the treatments.

Response:

3) In the case of Figure 1, the 2.1 LED module is mentioned, and this figure is used to verify the mentioned image.

4) Table 1 contains studies conducted in various fields of references. Accordingly, the temperature conditions are different according to the microorganism used, and thus it shows that the temperature conditions were performed in various ranges such as low, medium, and high temperatures. This study measured the experimental conditions without a separate setting for temperature to reflect the application of visible light LEDs in the general environment and described them in the results.

The temperature rise can be variously considered. After the microbial experiment, sterilization was carried out at high temperature (e.g., over 100 °C) in the autoclave. The heat generation of the LED module in this study was not considered to have a significant effect on sterilization. Rather, we judged that a temperature range that affects microbial growth could have been formed. Therefore, it was considered best to reduce the temperature as much as possible.

5) The authors have the same opinion as the commenter with regards to the number of repeated experiments in Table 3. In particular, as the source of pollution called microorganisms grows, there are many variables; therefore, the number of repetitions should be as high as possible as possible. In the case of UV-C with a short LED irradiation time (the longest time was about 11 min), the treatment could be repeated four times. However, some cases took more than one hour for UV-A or 405 nm irradiation. In such cases, only a low number of repetitions (on the same day) were possible to minimize the impact of proceeding over other days when distributing and applying strains for microbial identification work. In addition, the longer the microorganisms prepared for the experiment were stored, the more likely they were to die; therefore, few repetitions could be carried out in the long-term irradiation cases.

6) In this experiment, there were no separate statistical analyses because the sterilization rate according to the irradiation time was confirmed for the sterilization method (depending on LED-wavelength). The control case was used to calculate the reduction rate, but it is not used to calculate the survival rate. The reason why the control case results are included is to compare how much microorganisms diffuse according to wavelength.

Results:

7) In this section also, the figure captions are incomplete and lack important details.

8) Temperature in the bench is increasing of almost 5°C after each sterilization process, this seems important. There is a 6°C increase in temperature at 370 and 385 nm, and a 5°C increase at 405 nm. This may create problems for the application in real life. Was this measured more than once?  In addition, the authors state that the surface increase is low at 275 nm compared to the other wavelength, is it lower, but it is still a 4°C increase, or 22%. This does not seem insignificant.

9) The n should be added in the figure captions.

10) Line 286-287 – this belongs in the discussion. Same lines 296-297, 306-307 and 315-316.

11) Figure 7 needs a Y axis title. What is the data presented, mean ± standard error or standard deviation? This needs to be in the figure caption, with the n.

Response:

8) The sterilization experiment was conducted inside a clean bench, and the temperature was measured inside the bench and on the surface of the agar plate. Regarding the temperature increase mentioned in the results, 370, 385, and 405 nm were simultaneously conducted, and the temperature increase was shown as the result. 275 nm was the result of the solo performance, showing an increase of 1-2 °C. Considering this, because three modules were operated at the same time, it seems appropriate that an increase of 5 °C was made in the result of 370, 385, and 405 nm.

9) The n values are shown in Table 3.

10) As per your comment, this information has been summarized in the results and these lines have been added to the discussion.

11) The Y-axis title was omitted to make the picture visible. The title is the inactivation rate, as shown in Figure 7 (only % is indicated on the Y-axis).

Discussion and conclusions:

12) Globally, I think the discussion needs to be longer with comparison to the literature. There is a need to discuss the distance of sterilization (light to surface), this seem to be important in the introduction but is barely discussed and no comparison is made with literature.

13) Line 403: I do not agree with that statement, blue light, and visible light with wavelength close to the UV range are known to affect the human eye negatively. I agree that the effects of these wavelengths are much less important than the effects of UVC, however it would be false to assume they do not cause any damage to human. Hence, stating that they could be used in spaces with people in not appropriate. In this regard, I can not support the statement (2) of the conclusion.

14) Because aims were unclear in the introduction, I think the authors should rewrite them clearly and modify the discussion accordingly.

Response:

12) Most previous research has confirmed the sterilization effect at very short distance. A distance of 15 cm is the minimum for applications in the general environment, but also for use on furniture or household goods in homes. When applied to general lighting, the sterilization effect should be verified at a longer distance. This study verifies the sterilization effect of visible LED compared to UV-C and UV-A. The results also examine the applicability to general life because it is energy efficient compared to UV-A (lines 360-368).

13) The information was mentioned in the discussion, but we agree that supporting content is missing. References [29, 30] below were added and further described. Ref [29] evaluated that the 405 nm has sterilization capabilities, but is safe for human exposure when used within the appropriate range of irradiance. In [30], unlike UV rays, which cause deterioration and damage to ordinary materials, 405 nm belongs to the visible light spectrum and was found not to cause deterioration-like damage. For this part, a reference was added by describing it in the discussion part.

  1. International Commission on Non-Ionizing Radiation Protection. (2004). Guidelines on limits of exposure to ultraviolet radiation of wavelengths between 180 nm and 400 nm (incoherent optical radiation). Health Physics, 87(2), 171-186.
  2. Maclean, M., McKenzie, K., Anderson, J. G., Gettinby, G., & MacGregor, S. J. (2014). 405 nm light technology for the inactivation of pathogens and its potential role for environmental disinfection and infection control. Journal of Hospital Infection, 88(1), 1-11.

14) We have reviewed the Discussion and included lines 369-375 to ensure clarity. The relevant text is included below.

“The sterilization effect of the 405 nm visible light LED was determined as follows. The k value was derived using the microorganism and wavelength experimental results, and 275 nm (UV-C) was similar to that previously shown [3]. It was assumed that the validity of the experiment in [3] was verified. The k value for UV-A, visible light conducted in the same way was considered to be reliable. Although the sterilization effect of UV-A is clearly lower than that of UV-C, it was considered because it can be used in a living environment for safety and applicability.”

Reviewer 2 Report

The authors describe an article entitled “Wavelength-based Inactivation Effects of Light-emitting Diode Module on Indoor Microorganisms”. The topic of the manuscript is interesting, and the manuscript constitutes an interesting research article concerning the surface sterilization by mean of LEDs.

The work is well-written and a well-constructed introduction has been established by the authors. Sufficient spectra and figures are included in the manuscript for comprehension and clarity. Interesting and convincing results are also presented in this work. Overall, I think that this is a manuscript that I recommend for publication after inclusion of minor revisions.

1) A series of LEDs have been used for the experiments. However, choice of these wavelengths are not justified. Is it common wavelengths in this field ?

2) At present, LEDs emitting at 405 nm are the cheapest ones due to their wide use in 3D printing. Considering that a high sterilization efficiency is obtained at 405 nm, with low energy consumption, the fact that LEDs@405 nm are really cheap should be mentioned.

3) Figure 7 is in black and white colors. Please introduce colors in order to render the figure more attractive.

4) “ml’ should be replaced by “mL”.

Author Response

We appreciate your kind comments. We have carefully read and responded to each comment and revised our manuscript.

The authors describe an article entitled “Wavelength-based Inactivation Effects of Light-emitting Diode Module on Indoor Microorganisms”. The topic of the manuscript is interesting, and the manuscript constitutes an interesting research article concerning the surface sterilization by mean of LEDs.

The work is well-written and a well-constructed introduction has been established by the authors. Sufficient spectra and figures are included in the manuscript for comprehension and clarity. Interesting and convincing results are also presented in this work. Overall, I think that this is a manuscript that I recommend for publication after inclusion of minor revisions.

  • A series of LEDs have been used for the experiments. However, choice of these wavelengths are not justified. Is it common wavelengths in this field ?

Response: In the field of environmental engineering, microbial sterilization is mainly applied to the inside of air handling units or the upper part of a room using a UV-C wavelength (200-280 nm). Within the UV wavelengths, UV-C is known to be particularly effective for sterilizing microorganisms; however, there are many restrictions on its use because it is harmful to humans. UV-A wavelengths such as 360-390 nm are also used for sterilization. To verify and compare the sterilization effect of 405 nm, the wavelength of the UV-C and UV-A regions, which are representative wavelengths, were selected. The related information was described in reference [1] and mentioned in the Introduction.

  • At present, LEDs emitting at 405 nm are the cheapest ones due to their wide use in 3D printing. Considering that a high sterilization efficiency is obtained at 405 nm, with low energy consumption, the fact that LEDs@405 nm are really cheap should be mentioned.

Response: UV LED being expensive is explained in the Introduction; however, I agree with the comment that 405 nm LEDs being cheap is missing. I have added this to the information to lines 402-403: “In addition, 405 nm LEDs are less expensive than UV LEDs and can be freely produced in various forms along with 3D prints.”

  • Figure 7 is in black and white colors. Please introduce colors in order to render the figure more attractive.

Response: As suggested, I have added color to the figure while retaining a pattern to allow for greyscale printing and accessibility.

  • “ml’ should be replaced by “mL”.

Response: All instances of “ml” in the manuscript have been revised to “mL”.

Reviewer 3 Report

The manuscript reports the the sterilization effects on four kinds of representative bacteria and mold by using LED modules (with wavelengths of 275, 370, 385, and 405 nm). The authors demonstrated that the sterilization effect of 275 nm has no significant difference in the sterilization effects of 370 and 385nm (UV-A), and 405 nm (visible light) with longer exposure.  The manuscript should be published in Internaional Journal of Environmental Research and Public Health after authors amend some minor items listed below.

1. The authors should modify the abstract to highlight the research focus-visible light sterilization is effective and can be used in a general living environment.

2. The introduction should be shortened appropriately. Because the introduction is too long, the main body of the research content begins to appear after page 5.

3. The author needs to indicate the vertical axis scale in Figure 2a.

4. The line spacing becomes smaller from page 9.

5. Why theB. subtilis morphology corresponding to 275nm in Table 4 seems to be very different from that corresponding to other wavelengths.

6. The lines in Figure 3a are not easy to distinguish, please modify them to lines  and error bar with different colors.

7. Even the visible light with 405 nm can cause harm to human eyes, and the author needs to clarify this point.

Author Response

We appreciate your kind comments. We have carefully read and responded to each comment and revised our manuscript.

The manuscript reports the the sterilization effects on four kinds of representative bacteria and mold by using LED modules (with wavelengths of 275, 370, 385, and 405 nm). The authors demonstrated that the sterilization effect of 275 nm has no significant difference in the sterilization effects of 370 and 385nm (UV-A), and 405 nm (visible light) with longer exposure.  The manuscript should be published in Internaional Journal of Environmental Research and Public Health after authors amend some minor items listed below.

1) The authors should modify the abstract to highlight the research focus-visible light sterilization is effective and can be used in a general living environment.

Response: We have reviewed the abstract and added a sentence stating that 405 nm can be used in general living environments.

2) The introduction should be shortened appropriately. Because the introduction is too long, the main body of the research content begins to appear after page 5.

Response: Some information unrelated to main stream in the Introduction has been removed and rearranged, as per your comment.

References [18, 23-26] were summarized in Table 1 at the end of the Introduction, and information repeated in the manuscript has been removed.

3) The author needs to indicate the vertical axis scale in Figure 2a.

Response: The vertical axis scale in Figure 2a has been revised.

4) The line spacing becomes smaller from page 9.

Response: I have revised the line spacing of the manuscript.

5) Why the B. subtilis morphology corresponding to 275nm in Table 4 seems to be very different from that corresponding to other wavelengths.

Response: It is appropriate to show the morphology of B. subtilis as 275 nm in Table 4. At other wavelengths, the incubation time was checked incorrectly and incubated for about 10 h longer than the scheduled time.(miss check) However, the bacteria did not die. The colony spread further and could be counted by eye, which resulted in a different appearance.

  1. The lines in Figure 3a are not easy to distinguish, please modify them to lines  and error bar with different colors.

Response: Not only Figure 3(a) but also Figure 4, 5, and 6(a) have been changed to make it easier to distinguish between the lines.

  1. Even the visible light with 405 nm can cause harm to human eyes, and the author needs to clarify this point.

Response: The information was mentioned in the discussion, but we agree that supporting content is missing. References [29, 30] below were added and further described. Ref [29] evaluated that the 405 nm has sterilization capabilities but is safe for human exposure when used within the appropriate range of irradiance. In [30], unlike UV rays, which cause deterioration and damage to ordinary materials, 405 nm belongs to the visible light spectrum and was found not to cause deterioration-like damage. For this part, a reference was added by describing it in the discussion part.

  1. International Commission on Non-Ionizing Radiation Protection. (2004). Guidelines on limits of exposure to ultraviolet radiation of wavelengths between 180 nm and 400 nm (incoherent optical radiation). Health Physics, 87(2), 171-186.
  2. Maclean, M., McKenzie, K., Anderson, J. G., Gettinby, G., & MacGregor, S. J. (2014). 405 nm light technology for the inactivation of pathogens and its potential role for environmental disinfection and infection control. Journal of Hospital Infection, 88(1), 1-11.

Round 2

Reviewer 1 Report

Thank you for replying to all my comments. I still wish to address some concerns with the manuscript, that I really think will improve the future direction of the research and applications of the findings.

1. I understand that future applications will be assessed in other related research, and understand the temperature issue, which I think you explained well in the response to my comments. I wish this was explained as well in the manuscript (your response to #8). It is not mentioned in the methodology that all exposures were conducted simultaneously, this is an important details and should be added, not just in the discussion.

2. I really appreciate the modifications in the introduction. It helps the reader understand the previous studies in relation to yours.

3. I understand the explanation for the number of replicates obtained, however as a biologist working with various organisms, the duration of irradiation is fairly short compared to many other irradiation/exposures in my field. I do believe that this type of experiment can be replicated more than it was in this study. I strongly suggest that the authors think about how they could do more replicates in their future research.

4. I still believe statistical analysis would be beneficial to improve this manuscript and help make stronger conclusions. The authors could use ANOVA to compare the effect of the duration of irradiation on the reduction rate of each wavelength. And also to compare reduction rate of the wavelengths for each microorganism.

5. I still am not convinced that the visible light wavelength (405 nm) should be suggested in a living environment for long term use. There is more research than the ones provided (29, 30), demonstrating the effects of this wavelength on human skin and eyes: Lawrence et al., Scientific reports, (2018) 8:12722, DOI:10.1038/s41598-018-30738-6 and Argun et al., Eye (2014) 28, 752–760. I think more research still needs to be conducted on the safety of these wavelengths on the human eye and skin (in regards to distance and duration of exposure) before suggesting a long term use in a living environment. The authors needs to be careful in their concluding statement.

Minor comments: 

Figure captions are still incomplete, the n should be present in the figure and not in a separate table. For an example of complete figure caption, I suggest the authors look at their reference : Murdoch et al., 2012. This is also a good reference for statistic analysis.

Line 206: "The microorganisms 205 were diluted in a lyophilized strain in a NaCl" -> Should this say: diluted from a lyophilized strain? 

Line 207: 10-4 and 10-3 dilution, the -3 and -4 should be in superscript.

Author Response

Thank you for replying to all my comments. I still wish to address some concerns with the manuscript, that I really think will improve the future direction of the research and applications of the findings.

  1. I understand that future applications will be assessed in other related research, and understand the temperature issue, which I think you explained well in the response to my comments. I wish this was explained as well in the manuscript (your response to #8). It is not mentioned in the methodology that all exposures were conducted simultaneously, this is an important details and should be added, not just in the discussion.

Response:

I mentioned and revised including your comments in methods.

  1. I really appreciate the modifications in the introduction. It helps the reader understand the previous studies in relation to yours.

Response:

Thank you for your comments.

  1. I understand the explanation for the number of replicates obtained, however as a biologist working with various organisms, the duration of irradiation is fairly short compared to many other irradiation/exposures in my field. I do believe that this type of experiment can be replicated more than it was in this study. I strongly suggest that the authors think about how they could do more replicates in their future research.

Response:

Of course, I think there are many variables to deal with in future research, such as irradiation/exposure time and distance etc. I’ll consider your comments(more replicates) in future experiments.

  1. I still believe statistical analysis would be beneficial to improve this manuscript and help make stronger conclusions. The authors could use ANOVA to compare the effect of the duration of irradiation on the reduction rate of each wavelength. And also to compare reduction rate of the wavelengths for each microorganism.

Response:

The statistical analysis results are revised in manuscript. A one-way ANOVA comparison between irradiation time-reduction rates was calculated. In the ANOVA analysis of the reduction rate according to the irradiation time, the p value was 0.09 only at 370 nm among the experiments on Penicilium, so there was no significant difference. In the other cases, the p value was less than 0.05, indicating a significant difference at irradiation time. ANOVA analysis of the reduction rate of the wavelengths for each microorganism is difficult to compare because the irradiation time is different. Since it was difficult to compare the whole at once, only the ANOVA analysis of the reduction rate according to the aforementioned irradiation time was performed.

  1. I still am not convinced that the visible light wavelength (405 nm) should be suggested in a living environment for long term use. There is more research than the ones provided (29, 30), demonstrating the effects of this wavelength on human skin and eyes: Lawrence et al., Scientific reports, (2018)8:12722, DOI:10.1038/s41598-018-30738-6 and Argun et al., Eye(2014) 28, 752–760. I think more research still needs to be conducted on the safety of these wavelengths on the human eye and skin (in regards to distance and duration of exposure) before suggesting a long term use in a living environment. The authors needs to be careful in their concluding statement.

Response:

The references you mentioned have been reviewed. I agree with you that more research is still needed on safety of theses wavelengths on the human eye and skin (in regards to distance and duration of exposure). At the conclusion, it seems to have been stated with the conclusion that “safety was guaranteed”. Therefore, “more research on the exposure-safety of wavelengths should be conducted and carefully considered”, the conclusion was revised.

Minor comments: 

  1. a) Figure captions are still incomplete, the n should be present in the figure and not in a separate table. For an example of complete figure caption, I suggest the authors look at their reference : Murdoch et al., 2012. This is also a good reference for statistic analysis.
  2. b) Line 206: "The microorganisms 205 were diluted in a lyophilized strain in a NaCl" -> Should this say: diluted froma lyophilized strain? 
  3. c) Line 207: 10-4 and 10-3 dilution, the -3 and -4 should be in superscript.

Response:

  1. a) I represent the n in Figure 7. And all of Figure captions are revised with the reference : Murdoch et al., 2012.
  2. b) Line 206: I revised the sentence.
  3. c) Line 207: I revised the superscript.